# Determinants of non-communicable disease management among support staff in Putrajaya, Malaysia: The mediating role of attitude

Azrin Shah Abu Bakar[1], Haliza Abdul Rahman[1,2]*, Ahmad Iqmer Nashriq Mohd Nazan[3]

1 Department of Environmental and Occupational Health, Faculty of Medicine and Health Sciences, Universiti Putra Malaysia, Serdang, Selangor, Malaysia, 2 Institute for Social Science Studies, Universiti Putra Malaysia, Serdang, Selangor, Malaysia, 3 Department of Community Health, Faculty of Medicine and Health Sciences, Universiti Putra Malaysia, Serdang, Selangor, Malaysia

* dr.haliza@upm.edu.my

## Abstract

Non-communicable disease (NCD) is a major global health issue that contributes to morbidity and mortality problems across countries, including Malaysia. Individuals and social environment are increasingly recognized as critical factors that impact NCDs. Low-income households are a potentially vulnerable group in managing chronic diseases. Therefore, this study aims to examine the factors that influence the management of NCD. In addition, this study examines the role of attitude on the relationship between intrapersonal factor (knowledge) and NCD management. This study employs a cross-sectional survey of 200 support staff with NCD at selected ministries in Putrajaya, Malaysia. The data collected included socio-demographic information, knowledge of NCDs, attitudes towards NCDs, and the Chronic Illness Resources Survey (CIRS). Statistical Package of Social Sciences (SPSS) version 27 and Analysis of Moment Structures (AMOS) version 24 were employed to analyze the data through descriptive analysis and Structural Equation Model (SEM). The path analysis revealed that attitude, community, and societal factors significantly influence NCD management. Analysis of mediating effects indicated that attitude served as a significant mediator in the relationship between intrapersonal factor (knowledge) and NCD management. Hence, these findings provide a better understanding of influences and mediating effects, highlighting the need for tailored interventions to address this issue to improve NCD management.

## Introduction

Globalization, urbanization, technological advancements, and demographic (population aging) have contributed significantly to the increase in Non-Communicable Diseases (NCDs) in Malaysia. Despite extensive government initiatives, including health

**Data availability statement:** The dataset contains sensitive human participant data involving individuals with non-communicable diseases; therefore, legal and ethical considerations prevent public disclosure of the dataset to protect participant confidentiality. For all data requests, please contact the corresponding author (dr. haliza@upm.edu.my) together with the Ethics Committee of Universiti Putra Malaysia (jke-upm@upm.edu.my).

**Funding:** The author(s) received no specific funding for this work.

**Competing interests:** The authors have declared that no competing interests exist.

programs, public campaigns, screening activities, and community-based health promotion efforts related to NCDs, the prevalence of NCDs in Malaysia continues to increase over the years.

Based on the National Health and Morbidity Survey (NHMS) in Malaysia, the prevalence of diabetes, hypertension, hypercholesterolemia, overweight, and obesity remains high among the population [1]. Moreover, NHMS from 2015 to 2019 reported that there is an increasing trend of NCDs such as diabetes and hypertension among the low-income groups compared to other income groups [2,3]. These findings highlight the persistent and growing burden of NCDs among the population, particularly among socioeconomically disadvantaged groups.

Additionally, the increase in NCDs has significant impacts on individual health, quality of life (QoL), and workforce productivity, contributing to increased absenteeism, reduced work performance, and a financial burden on the country [4]. Indeed, underprivileged populations often face limited access to quality healthcare services and facilities, placing them at high risk for NCDs [5]. Moreover, they may also experience loss of family income due to treatment costs and healthcare expenses, or bear the financial burden of NCD [6]. Thus, low-income earners are disproportionately affected by the growing burden of NCDs.

Addressing NCD requires a comprehensive, efficient, and cost-effective approach to improve health and well-being. Additionally, understanding the factors associated with NCD is crucial for developing health interventions for this disadvantaged group. NCD management is one approach to improve the QoL and prevent complications of the disease. It is also considered an effective and low-cost initiative for preventing and managing NCDs [7,8]. This is consistent with a previous study, which reported a significant relationship between QoL, general health, and management of NCD [9]. Therefore, NCD management is one of the best approaches to improving the QoL and well-being.

Broadly, NCD is a complex condition that requires engagement from multiple contributors, including individual, social, community, and societal factors [10]. This is mainly attributed to the fact that personal, social, organizational, community, and societal factors can positively or negatively impact an individual's influence on behavior. Therefore, effective strategies for NCD management require multi-sectoral collaboration and action. This is aligned with the National Strategic Plan for Non-Communicable Diseases (NSP-NCD) 2016–2025, which provides a comprehensive framework to strengthen strategies for the prevention and control of non-communicable diseases in Malaysia, with an emphasis on a whole-of-government and whole-of-society approach.

In this study, Socio-Ecological Model is one of the most commonly utilized theories formulated by McLeroy et al. (1988) to explain environmental causes of health behavior and health promotion. This includes identifying interventions that offer five levels of influence specific to health behavior, such as intrapersonal, interpersonal, organizational, community, and societal factors [11]. In addition, the Socio-Ecological Model can provide a comprehensive understanding of the various influences on health outcomes and on reducing health inequality, especially among low-income groups [12].

In line with this, attitude plays a crucial role in shaping human behavior and can be considered a mediator of preventive health behaviors. Attitude is the evaluation, feelings, and tendencies towards objects, events, groups, or symbols that significantly influence individuals to engage in actions that promote health and prevent disease [13]. Attitude can also drive motivation to initiate health-related behavior actions. Essentially, individuals who have a positive attitude can influence engagement to behave towards the management of NCDs. In contrast, a negative attitude implies a low attitude evaluation that can influence and hinder the development of good NCD management. This suggests that attitude is a psychological factor in shaping decisions that encourage a person to engage in preventive behavior. Therefore, an important emphasis on understanding predictors and mediating attitudes should be studied in determining behavior related to NCD management, as the nature of this disease is characterized by a long duration, slow progression, incurability, and lifelong conditions. In other words, continuous management of NCDs needs to be emphasized. Nevertheless, there remains a limited number of studies on the management of NCDs among low-income households. Most studies on NCD among low-income earners have more focus on several factors such as demographic, socio-economic and health characteristics [6], prevalence of NCDs, socio-demographic and socio-economic factors [14,15], socio-demographic factors [16], demographic and socio-economic factors [17], health status and living needs [18], health, mental health, health and nutritional behaviors [19], and QoL [20]. Furthermore, there is a lack of local studies examining the mediating role of attitude within the framework of the socio-ecological model to assess how intrapersonal factors, such as knowledge, influence NCD management.

In this study, attitude functions as a mediator of behavioral processes, which is conceptualized as a psychological mechanism through which socio-ecological factors (i.e., intrapersonal) are translated into NCD management behaviors. The theoretical model is shown in Fig 1. The Knowledge–Attitude–Practice (KAP) model, along with other theoretical models were used to understand health-related behaviors associated with health issues. The KAP model posits that knowledge shapes attitudes, which in turn influence health-related practices [21–24]. In other words, knowledge (awareness and understanding) shapes attitudes (feelings and beliefs), which subsequently drive health-related actions. Therefore, examining the mediating role of attitude provides a clearer theoretical understanding of how socio-ecological factors (i.e., intrapersonal) influence NCD management, particularly among low-income populations.

## Materials and methods

### Study design and study location

A cross-sectional study was conducted among support staff at selected ministries in Putrajaya. Putrajaya is located approximately 25 kilometers south of the capital, Kuala Lumpur [25,26]. It is the administrative center of the Malaysian government, replacing Kuala Lumpur as the Government Administrative Center to reduce congestion in the area and to address the shortage of government land needed to accommodate the increasing demand for office space [27]. The establishment of a new administrative center at a dedicated site has enabled the development of a well-planned urban center equipped with modern facilities and technology to enhance the efficiency and productivity of government administration, while also ensuring a higher quality of urban life and environment. Additionally, Putrajaya is a planned city developed based on two main themes, namely a garden city and a smart city. It comprises government ministry complexes, residential areas for civil servants, lakes, and green spaces to promote a healthy urban environment [26–29]. Hence, this location was selected, as it is the Federal Government Administration Center, which represents the largest number of civil servants in Malaysia.

### Sampling technique and sample

Two-stage cluster sampling method was employed. In the first stage, eight out of 24 ministries in Putrajaya were randomly selected to represent the ministries in Putrajaya. In the second stage, eligible support staff were selected using computer-generated simple random sampling based on name lists obtained from the Human Resources Division of the selected ministries. Only permanent support staff aged 18–60 years with non-communicable diseases were included, while contract or part-time staff and those who declined participation were excluded.

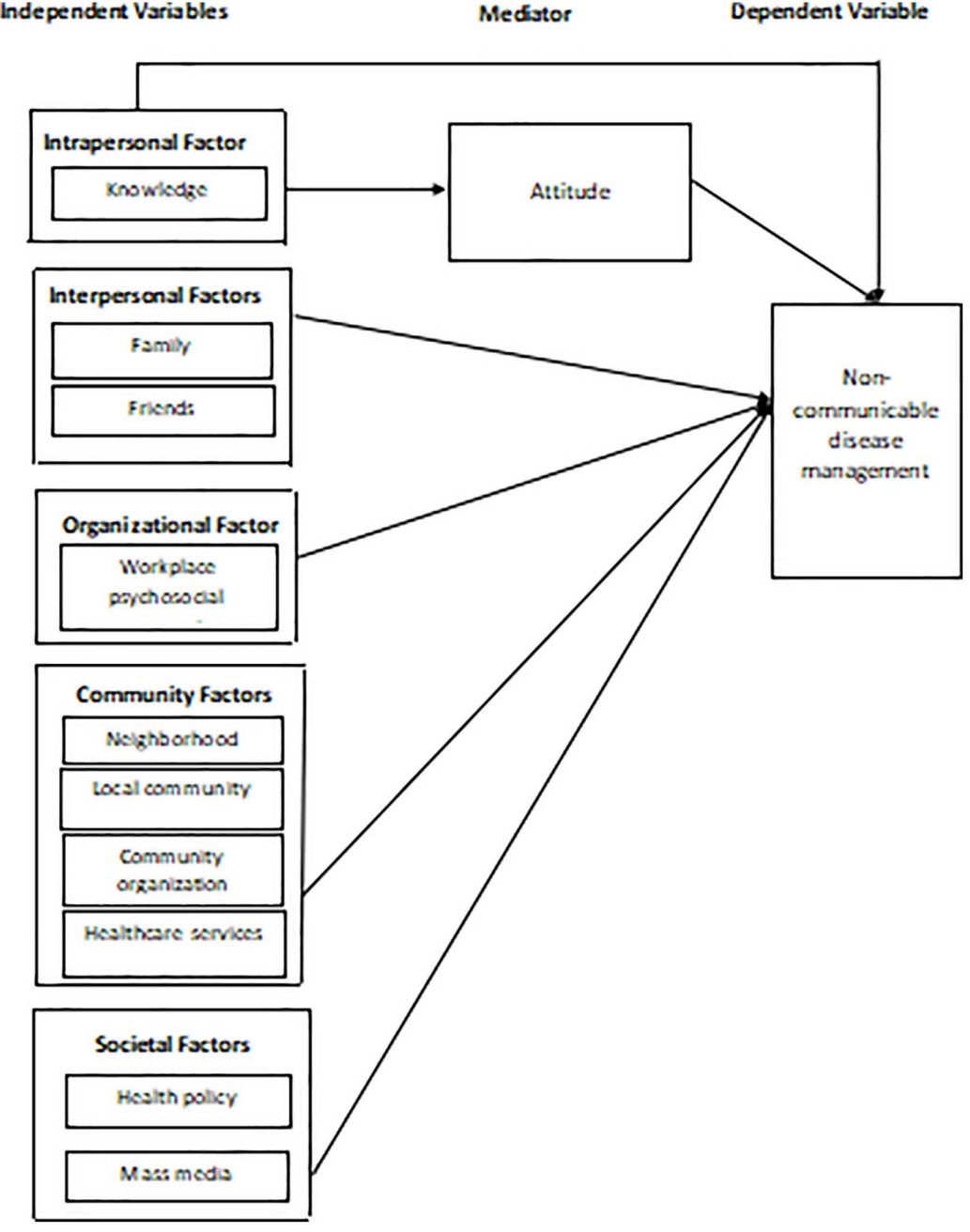

**Fig 1. The theoretical model.**

## Sample size

Sample size based on an online calculator for Structural Equation Modeling (SEM) of Soper, D.S. (2020) by setting the anticipated effect size $f^2 = 0.3$, the desired p-value was at 0.05, the desired statistical power level at 80%, the number of observed variables was 94, the number of latent variables was 12, and the number of minimum sample sizes suggested is 200 [30,31,32]. According to Kline (2010), the sample size for SEM should be in the range of 100–200 subjects, which is

considered a medium-sized sample, while a sample size above 200 is considered large. Hence, this study considered 200 samples, adjusted for a 20% non-response rate, and the minimum sample size required was 240 [33].

## Instruments

A set of questionnaires was distributed to respondents who fulfilled the inclusion criteria via an online survey using a Google Form link. This study instrument comprises four sections, which include socio-demographic information, knowledge of NCDs, attitudes towards NCDs, and the Chronic Illness Resources Survey (CIRS). The questionnaire was adapted from previous studies [31,32]. Section A consists of socio-demographic information, including age, gender, race, marital status, educational level, and monthly household income, smoking status, Body Mass Index (BMI) status, and taking medications status. Section B assesses the respondents' knowledge of NCD. The knowledge questions cover the understanding of NCD, including the major NCD such as cardiovascular disease, stroke, and hypertension, diabetes mellitus, and chronic obstructive pulmonary disease, as well as risk factors, complications, and disease management. Respondents were instructed to respond with 'yes,' 'no,' and 'do not know.' Knowledge levels were classified as poor (0–50%), moderate (51–75%), and good (76–100%). It had excellent internal consistency reliability (Cronbach's α=0.961). Section C consists of 15 items assessing attitudes towards NCD, including balanced diet, physical activity, smoking, salt intake, medication intake, and treatment checks. The question was evaluated using a 5-point Likert scale ranging from 'Strongly Agree,' 'Agree,' 'Not Sure', 'Disagree,' and 'Strongly Disagree'. The total score was divided into three categories as follows: poor attitude (scores 0–50%), moderate attitude (scores 51–75%), and good attitude (scores 76–100%). In this study, the survey revealed good internal consistency reliability (Cronbach's α=0.898). Section D was the final section of the questionnaire containing 65 items for the CIRS to assess socio-ecological factors, divided into five sub-scales that measure interpersonal, organizational, community, societal, and disease management. The questionnaire is based on a 5-point Likert scale: 'Not at all,' 'A little,' 'Moderate,' 'Quite a lot,' and 'A great deal.' The total score can be categorized as follows: less than 60% is considered low, 60% to 80% is considered moderate, and 80% and above is considered high. The Cronbach's Alpha for the dimensions of the questionnaire ranged from 0.70 to 0.90, demonstrating good reliability.

In addition, the Average Variance Extracted (AVE) values ranged from 0.463 to 0.730 and Composite Reliability (CR) values ranged from 0.778 to 0.955. Despite AVE values being less than 0.50, it is still acceptable provided CR values are greater than 0.6 [34]. This is consistent with criteria established by previous researchers, who suggested that AVE values above 0.45 and CR values above 0.60 indicate adequate convergent validity [35,36]. Therefore, convergent validity was achieved for all constructs. Additionally, the square roots of AVE values for all sub-constructs were higher than the inter-construct correlations, confirming that discriminant validity was achieved. Therefore, it demonstrates satisfactory reliability and construct validity, meeting the prerequisites for SEM analysis.

## Ethics considerations

Ethical approval for this study was obtained from the Ethics Committee of Universiti Putra Malaysia (JKEUPM-2024–301). Prior to data collection, all respondents were provided with written informed consent that explained their participation in the study was voluntary and outlined the study's purpose, risks, and benefits. The confidentiality of the information was maintained in the strictest confidence, and no names or identifying information were recorded to ensure participant anonymity during data analysis.

## Data analysis

Data analysis was performed using the Statistical Package of Social Sciences (SPSS) version 27 and the Analysis of Moment Structure (AMOS) version 24. Meanwhile, descriptive analysis was performed using frequencies and percentages. Following this, Structure Equation Model (SEM) analysis was employed to examine the interrelationship between

the constructs in the hypothesized model. Subsequently, a test of the mediation effect was performed through a bootstrap analysis. The significance level was set at p < 0.05.

## Results

### Socio-demographic characteristics

The socio-demographic characteristics of the respondents are summarized in Table 1. The majority of the respondents were aged between 40 and 49 years old (67.5%), female (69.5%), Malay (95.0%), married (73.5%), household income of the respondents ranges between RM3501 and RM4000 (51.0%), and completed a higher educational level of at STPM/ Matriculation/ Diploma and SPM/Certificate (69.0%). Approximately 43.0% of the respondents had high cholesterol, while 41.0% of them were obese. The majority of them (93.0%) were non-smokers. More than half of them (56.0%) were taking medications regularly.

### Structural equation modeling analysis

Structural Equation Modeling (SEM) was employed to assess the hypotheses of the study to determine factors influencing NCD management and the mediating role of attitude in these relationships. Using this method, hypotheses can explain relationships among multiple variables and the degree of their correlations with the data obtained simultaneously. Subsequently, the model fit was determined based on the Goodness-of-Fit (GOF) [37]. The model fit was assessed using four fit indices: relative Chi-square ($\chi^2$/df) ≤ 5.0; root mean square error of approximation (RMSEA) ≤ 0.08; comparative fit index (CFI) ≥ 0.90; and incremental fit index (IFI) ≥ 0.90. Hair et al. (2010) recommended that three to four indices, including one absolute index and one incremental index, meet the criteria, which provides adequate evidence of model fit [25].

Based on SEM, the resulting structural model fits indicated the data fit the model with [ChiSq/df = 1.693, P = 0.000, CFI = 0.920, TLI = 0.912, RMSEA = 0.059]. The value of the relative chi-square ($\chi^2$/df) was well below 5, indicating an acceptable fit between the proposed measurement model and the collected data. Other fit indices, namely CFI and TLI values, were above the minimum value of .90, and the RMSEA value was below the cut-off value of .08 (RMSEA = .050). Therefore, the analysis of the SEM indicated that the overall value of the fit index reached acceptable levels and confirmed that the structural model was a good fit.

### Structural model

In this study, structural model analysis was employed to conduct path analysis and evaluate the research hypotheses. Table 2 summarizes the results of the hypothesized path analysis of the structural model of this study. The influence of intrapersonal factors (knowledge), interpersonal factors (family and friends), organizational factors (workplace psychosocial support), community factors (neighborhood, local community, community organization, and healthcare services), societal factors (health policy of healthcare financing and mass media) and attitude on NCD management was assessed through path analysis.

The results indicated that attitude (β = 1.522, p < 0.001), community factors (β = −0.259, p < 0.001), and societal factors (β = 0.439, p < 0.001) had a significant effect on NCD management. This indicated that attitude, community factors, and societal factors significantly influence NCD management.

### Test of mediation effects

The bootstrapping procedure was conducted to evaluate mediation effects. Table 3 shows that there was no significant effect between intrapersonal factor (knowledge) and NCD management in direct model (Beta = −0.062, p = 0.314). Meanwhile, the result found there was a significant indirect effect of intrapersonal factor (knowledge) on NCD management through attitude by a 95% boot confidence interval (Beta = −0.077, p = 0.003) does not include zero. Thus, these

**Table 1. Socio-demographic characteristics of the respondents.**

| Socio-demographic Characteristics | Frequency (n) | Percentage (%) |
|---|---|---|
| **Age** | | |
| Below than 30 years old | 11 | 5.5 |
| 30 - 39 years old | 43 | 21.5 |
| 40 - 49 years old | 135 | 67.5 |
| 50 - 59 years old | 11 | 5.5 |
| **Gender** | | |
| Male | 61 | 30.5 |
| Female | 139 | 69.5 |
| **Ethnicity** | | |
| Malay | 190 | 95.0 |
| Chinese | 7 | 3.5 |
| Indian | 0 | 0.0 |
| Others | 3 | 1.5 |
| **Marital Status** | | |
| Single | 29 | 14.5 |
| Married | 147 | 73.5 |
| Widowed/Divorced/Separated | 24 | 12.0 |
| **Monthly Household Income** | | |
| RM1001 - RM1500 | 0 | 0.0 |
| RM1501 - RM2000 | 18 | 9.0 |
| RM2001 - RM2500 | 16 | 8.0 |
| RM2501 - RM3000 | 21 | 10.5 |
| RM3001 - RM3500 | 37 | 18.5 |
| RM3501 - RM4000 | 102 | 51.0 |
| More than RM4000 | 6 | 3.0 |
| **Educational Level** | | |
| Degree | 62 | 31.0 |
| STPM/ Matriculation/ Diploma | 69 | 34.5 |
| SPM/ Certificate | 69 | 34.5 |
| PT3/ PMR/ SRP | 0 | 0.0 |
| **Types of Non-Communicable Diseases** | | |
| Diabetes Mellitus | 55 | 27.5 |
| Heart Disease | 10 | 5.0 |
| Stroke | 3 | 1.5 |
| Chronic Obstructive Pulmonary Disease (COPD) | 4 | 2.0 |
| High Cholesterol | 86 | 43.0 |
| Cancer | 3 | 1.5 |
| Kidney Disease | 4 | 2.0 |
| **Body Mass Index (BMI)** | | |
| Underweight | 11 | 5.5 |
| Normal | 38 | 19.0 |
| Overweight | 69 | 34.5 |
| Obese | 82 | 41.0 |
| **Smoking status** | | |
| Yes | 14 | 7.0 |
| No | 186 | 93.0 |

*(Continued)*

**Table 1.** (Continued)

| Socio-demographic Characteristics | Frequency (n) | Percentage (%) |
|---|---|---|
| **Status of prescribed medication intake** | | |
| Yes | 112 | 56.0 |
| No | 88 | 44.0 |

**Table 2. Regression weight of the direct hypothesized model.**

| Construct | Path | Construct | Estimate | S.E. | C.R. | P |
|---|---|---|---|---|---|---|
| Non- Communicable Disease Management | <--- | Attitude | 1.522 | 0.282 | 5.402 | *** |
| Non- Communicable Disease Management | <--- | Intrapersonal Factor | −0.017 | 0.016 | −1.083 | 0.279 |
| Non- Communicable Disease Management | <--- | Interpersonal Factors | 0.176 | 0.139 | 1.264 | 0.206 |
| Non- Communicable Disease Management | <--- | Organizational Factors | 0.025 | 0.146 | 0.17 | 0.865 |
| Non- Communicable Disease Management | <--- | Community Factors | −0.259 | 0.057 | −4.562 | *** |
| Non- Communicable Disease Management | <--- | Societal Factors | 0.439 | 0.073 | 5.99 | *** |

SE: Standard Error; CR: Critical Ratio for regression weight; P: *** < 0.001.

**Table 3. Results of bootstrap analysis.**

| Model/Hypothesized Path | Beta | P | 95% CI BC | | Result |
|---|---|---|---|---|---|
| | | | Lower Bound | Upper Bound | |
| **Direct Model:** | | | | | |
| Intrapersonal Factor → Non-Communicable Disease Management | −0.062 | 0.314 | −0.155 | 0.035 | Not Significant |
| **Indirect Model:** | | | | | |
| Intrapersonal Factor → Attitude → Non-Communicable Disease Management | −0.077 | 0.003 | −0.154 | −0.036 | Significant |

findings indicated that attitude fully mediated the relationship between the intrapersonal factor (knowledge) and NCD management.

## Discussion

This study revealed that attitude, community, and social factors influence NCD management. Attitude was identified as the strongest influence on NCD management. This may be due to having a good attitude; individuals are confident in practice towards the management of chronic disease [38]. Therefore, when individuals have a positive attitude, it helps to improve their management of NCDs. Additionally, findings revealed that community factors (neighborhood, local community, community organization, and healthcare services) influence NCD management. It indicated that community factors effectively influence NCD management. One probable explanation is that the approach of health promotion programs applied in the community has increased their awareness of NCD. This result was consistent with previous studies, which have reported that community-based programs, such as Komuniti Sihat Perkasa Negara (KOSPEN), help address NCDs and prevent and control NCD risk factors [39]. Another study discovered that community engagement involves partnerships with local health organizations, such as community outreach programs and hosting health awareness events, which contribute to the overall well-being of the communities [40]. Therefore, the involvement of community health workers as part of the healthcare team is one of the effective efforts of the community program in increasing access to treatment and healthcare. This indicated that the community approach initiative helps ongoing healthcare by preventing and controlling NCD

risk within the community [41]. Another possible reason is that Putrajaya was designed with a neighborhood concept in mind, adopting a garden city concept that provides residents with neighborhood parks, including green spaces, gardens, and lakes, promoting physical activity (e.g., walking, jogging, and cycling). A previous study noted that open spaces in Putrajaya significantly contribute to health promotion [42]. A similar finding was also noted, highlighting that the presence of green urban spaces promotes healthy communities and healthy lifestyles [43]. Thus, it encourages a healthier lifestyle through community engagement in outdoor activities, fostering community bonds in shared recreational spaces. Furthermore, since the respondents live in an urban area, the majority of them have easy access to healthcare services. Urban areas make it easier for people to access healthcare services due to better public transportation options and a variety of clinic and hospital services, which influence the management of NCDs. In fact, individuals from relatively low-income earners prefer to choose services from their community, where they need only pay MYR 1 for a general outpatient consultation and MYR 5 for a specialist consultation [44]. This is consistent with the report by NHMS 2019, which revealed that 8.1% of B40 households in Malaysia received healthcare services as outpatients in the two weeks prior to consulting healthcare professionals [2]. Moreover, the government's role at the primary care level in NCD management has seen an increase in the roles of doctors, nurses, and medical assistants, where follow-up care is provided at many primary health clinics [45]. This is supported by [46], who mentioned that management in the primary care system provides services to the community for effective disease management and patient empowerment. This subsequently leads to better health status among NCD patients as well as the general community. Furthermore, healthcare services help prevent and treat health problems for the public, which in turn influences the population's overall health status. Therefore, community factors such as access to healthcare facilities, availability of recreational spaces, and community health programs had an effect on NCD management among support staff.

This study also revealed that societal factors (health policy of healthcare financing, and mass media) were identified as significant factors that significantly influenced NCD management. This could be a health policy in healthcare financing by the government, helping to protect them from NCDs. This aligns with the findings of the NHMS (2009–2023), which reported that government spending on health, particularly for low-income groups, reflects a strong public role in health financing for these groups. For example, PeKa B40 aims to improve access to healthcare services for low-income groups [47]. In addition, heavily subsidized by the government, providing affordable and accessible care to all citizens, especially low-income groups. Therefore, government healthcare financing plays a significant role in protecting populations, especially low-income groups. On the other hand, mass media also play a vital role in influencing NCD management. The existence of mass media provides numerous benefits to the public, making it easier for them to obtain, share, and exchange information, which in turn influences healthy behavior. The findings are consistent with prior research indicating that social media influences preventive behavior by increasing awareness and facilitating information exchange [48]. Another study emphasized that the use of mass media influences health behavior [49]. This implies that NCD management can be improved by enhancing attitude, community, and societal factors.

The current study revealed that attitude mediates the relationship between intrapersonal factor (knowledge) and NCD management. This study found that attitude fully mediates the relationship between the intrapersonal factor (knowledge) and NCD management. The present study highlights a significant indirect pathway by which an intrapersonal factor (knowledge) is associated with NCD management through attitude. This indicated that attitude helps regulate the relationship between intrapersonal factor (knowledge) and NCD management. In other words, intrapersonal factor (knowledge) lead to NCD management practices, driven by attitude. Furthermore, by increasing an individual's level of awareness, their building practices toward the prevention of NCD can be improved. The findings of this study were consistent with previous studies, which have proven attitude to be a mediator in controlling the relationship between exogenous variables and endogenous variables [50,51]. Hence, attitude serves as a mediator, influencing interpersonal factor in NCD management. Taken together, relevant stakeholders might better consider mediated relationships in a tailored intervention toward achieving NCD management.

 

## Conclusion

In summary, attitude, community, and societal factors significantly influence NCD management. Furthermore, attitude was found to fully mediate in the relationship between intrapersonal factor (knowledge) and NCD management. The contribution of this study is the identification of socio-ecological factors as significant determinants, as well as the role of attitude on mediating the relationship between intrapersonal factors (e.g., knowledge) and NCD management. Furthermore, this study provides new insights for relevant stakeholders, including policymakers, healthcare providers, and programme managers, by offering a better understanding of these influences and mediating effects in designing holistic interventions for the future. However, the study has limitations, such as the use of self-report questionnaires, which may have given false or incomplete responses. Additionally, the cross-sectional design makes it impossible to establish causality, which longitudinal studies could help determine. In addition, this study only utilized socio-ecological factors (e.g., intrapersonal, interpersonal, organizational, community, and societal). The assessment of other factors, such as health literacy, self-efficacy, and empowerment should be considered in this study, as they can contribute to a deeper understanding of NCD management. Lastly, future studies can also be conducted on larger samples by utilizing SEM analysis to confirm the good results and generalizability.

## Supporting information

**S1 File. Questionnaire.**
(DOCX)

**S2 File. Instrument.**
(DOCX)

## Author contributions

**Conceptualization:** Azrin Shah Abu Bakar, Ahmad Iqmer Nashriq Mohd Nazan.

**Investigation:** Azrin Shah Abu Bakar.

**Methodology:** Haliza Abdul Rahman, Azrin Shah Abu Bakar, Ahmad Iqmer Nashriq Mohd Nazan.

**Supervision:** Haliza Abdul Rahman, Ahmad Iqmer Nashriq Mohd Nazan.

**Validation:** Azrin Shah Abu Bakar, Ahmad Iqmer Nashriq Mohd Nazan.

**Writing – original draft:** Azrin Shah Abu Bakar.

**Writing – review & editing:** Haliza Abdul Rahman, Ahmad Iqmer Nashriq Mohd Nazan.

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
