## [Decision Letter · Decision Letter 0]

6 Oct 2025

Dear Dr. Abdul Rahman,

Thank you for submitting your manuscript to PLOS ONE. After careful consideration, we feel that it has merit but does not fully meet PLOS ONE’s publication criteria as it currently stands. Therefore, we invite you to submit a revised version of the manuscript that addresses the points raised during the review process.

We look forward to receiving your revised manuscript.

Kind regards,

Avanti Dey, PhD

Staff Editor

PLOS ONE

Journal Requirements:

Reviewers' comments:

Reviewer's Responses to Questions

**Comments to the Author**

1. Is the manuscript technically sound, and do the data support the conclusions?

Reviewer #1: Yes

2. Has the statistical analysis been performed appropriately and rigorously?

Reviewer #1: Yes

3. Have the authors made all data underlying the findings in their manuscript fully available?

Reviewer #1: No

4. Is the manuscript presented in an intelligible fashion and written in standard English?

Reviewer #1: No

Reviewer #1: The study titled “Mediating Role of Attitude on the Relationship Between Socio-Ecological Factors and Non-Communicable Disease Management among Support Staff in Putrajaya, Malaysia” addresses a critical public health concern. Non-communicable diseases (NCDs) remain a leading cause of mortality and morbidity globally, with disproportionate impacts on low-income and vulnerable populations.

By applying the Socio-Ecological Model (SEM), this research highlights the multi-layered determinants of health behavior — from individual to societal levels — and introduces attitude as a key psychological mediator that influences NCD management practices. The use of Structural Equation Modeling (SEM) strengthens the study's methodological rigor, and the focus on support staff in Putrajaya offers contextual insight into a population that is often underrepresented in health research.

This study may contribute to the empirical literature by explicitly testing the mediating role of attitude within a socio-ecological framework and provides actionable evidence for policymakers, healthcare providers, and program managers seeking to enhance NCD interventions through behaviorally informed strategies.

However, to improve the rigor of the study and to benefit the readers who are unfamiliar with the modelling strategies used, I attached the comments to manuscript that can be viewed to further improve the readability.

I would also suggest for English proofreading to improve the structure especially the results and discussion section.

.

Reviewer #1: **Yes:** Farah Ayuni ShafieFarah Ayuni ShafieFarah Ayuni ShafieFarah Ayuni Shafie

While revising your submission, please upload your figure files to the Preflight Analysis and Conversion Engine (PACE) digital diagnostic tool, https://pacev2.apexcovantage.com/. PACE helps ensure that figures meet PLOS requirements. To use PACE, you must first register as a user. Registration is free. Then, login and navigate to the UPLOAD tab, where you will find detailed instructions on how to use the tool. If you encounter any issues or have any questions when using PACE, please email PLOS at . PACE helps ensure that figures meet PLOS requirements. To use PACE, you must first register as a user. Registration is free. Then, login and navigate to the UPLOAD tab, where you will find detailed instructions on how to use the tool. If you encounter any issues or have any questions when using PACE, please email PLOS at . PACE helps ensure that figures meet PLOS requirements. To use PACE, you must first register as a user. Registration is free. Then, login and navigate to the UPLOAD tab, where you will find detailed instructions on how to use the tool. If you encounter any issues or have any questions when using PACE, please email PLOS at . PACE helps ensure that figures meet PLOS requirements. To use PACE, you must first register as a user. Registration is free. Then, login and navigate to the UPLOAD tab, where you will find detailed instructions on how to use the tool. If you encounter any issues or have any questions when using PACE, please email PLOS at figures@plos.org. Please note that Supporting Information files do not need this step.. Please note that Supporting Information files do not need this step.

---

## [Author Response · Author response to Decision Letter 1]

21 Nov 2025

I would like to express my gratitude to the reviewers for their helpful criticisms and comments, which have enabled us to improve the quality of this manuscript.

Please refer to the manuscript highlighted in blue for our responses to each of the reviewers’ comments.

Thank you.

---

## [Decision Letter · Decision Letter 1]

4 Jan 2026

Thank you for submitting your manuscript to PLOS ONE. After careful consideration, we feel that it has merit but does not fully meet PLOS ONE’s publication criteria as it currently stands. Therefore, we invite you to submit a revised version of the manuscript that addresses the points raised during the review process.

We look forward to receiving your revised manuscript.

Kind regards,

Vasudevan Ramachandran

Academic Editor

PLOS One

Journal Requirements:

Additional Editor Comments (if provided):

Dear Authors,

Please revise the manuscript according to the comments provided by the reviewers.

Reviewers' comments:

Reviewer's Responses to Questions

**Comments to the Author**

Reviewer #1: All comments have been addressed

Reviewer #2: (No Response)

2. Is the manuscript technically sound, and do the data support the conclusions?

Reviewer #1: Yes

Reviewer #2: Partly

3. Has the statistical analysis been performed appropriately and rigorously?

Reviewer #1: Yes

Reviewer #2: No

4. Have the authors made all data underlying the findings in their manuscript fully available?

Reviewer #1: Yes

Reviewer #2: Yes

5. Is the manuscript presented in an intelligible fashion and written in standard English?

Reviewer #1: Yes

Reviewer #2: No

Reviewer #1: (No Response)

Reviewer #2: This paper lack ground basis why there is a need to have additional attitude construct with SEM on the NCD management. Please see comments in the attached document.

.

Reviewer #1: **Yes:** Farah Ayuni ShafieFarah Ayuni ShafieFarah Ayuni ShafieFarah Ayuni Shafie

Reviewer #2: No

---

## [Author Response · Author response to Decision Letter 2]

29 Jan 2026

We appreciate your time in reviewing our paper and providing valuable comments. It was your valuable and insightful comments that led to possible improvements in the current version. The authors have carefully considered the comments and tried our best to address every one of them.

Below, we provide the point-by-point responses. All modifications in the manuscript have been highlighted in blue.

Sincerely,

Haliza Abdul Rahman, PhD

Associate professor

Department of Environmental and Occupational Health,

Faculty of Medicine and Health Sciences,

Universiti Putra Malaysia,

43400 Serdang, Selangor, Malaysia.

---

## [Decision Letter · Decision Letter 2]

5 Apr 2026

Determinants of Non-Communicable Disease Management: The Mediating Role of Attitude among Support Staff in Putrajaya, Malaysia

PONE-D-25-10747R2

Dear Dr. Haliza, Abdul Rahman

We’re pleased to inform you that your manuscript has been judged scientifically suitable for publication and will be formally accepted for publication once it meets all outstanding technical requirements.

Kind regards,

Vasudevan Ramachandran

Academic Editor

PLOS One

Additional Editor Comments (optional):

All the comments given by the reviewers were addressed by the authors and it is well organized now.

Reviewers' comments:

Reviewer's Responses to Questions

**Comments to the Author**

Reviewer #2: All comments have been addressed

2. Is the manuscript technically sound, and do the data support the conclusions?

Reviewer #2: Yes

3. Has the statistical analysis been performed appropriately and rigorously?

Reviewer #2: Yes

4. Have the authors made all data underlying the findings in their manuscript fully available?

Reviewer #2: Yes

5. Is the manuscript presented in an intelligible fashion and written in standard English?

Reviewer #2: Yes

Reviewer #2: Thank you very much for addressing most of the comments in the 1st review. Just need a bit of improvement for the paper to be published.

.

Reviewer #2: No

---

## [Editor Report · Acceptance letter]

PONE-D-25-10747R2

PLOS One

Dear Dr. Abdul Rahman,

I'm pleased to inform you that your manuscript has been deemed suitable for publication in PLOS One. Congratulations! Your manuscript is now being handed over to our production team.

Kind regards,

on behalf of

Professor Vasudevan Ramachandran

Academic Editor

PLOS One